# The Role of Cognitive Reserve in Post-Stroke Rehabilitation Outcomes: A Systematic Review

**DOI:** 10.3390/brainsci14111144

**Published:** 2024-11-15

**Authors:** Debora Bertoni, Stefania Bruni, Donatella Saviola, Antonio De Tanti, Cosimo Costantino

**Affiliations:** 1Centro Cardinal Ferrari, Via IV Novembre 21, 43012 Fontanellato, Italy; stefania.bruni.83@gmail.com (S.B.); donatella.saviola@kosgroup.com (D.S.); antonio.detanti@kosgroup.com (A.D.T.); 2Department of Medicine and Surgery, University of Parma, 43126 Parma, Italy; cosimo.costantino@unipr.it

**Keywords:** cognitive reserve, stroke, cognitive outcome, functional outcome, rehabilitation

## Abstract

**Background/Objectives**: Stroke remains a major cause of disability and death, with survivors facing significant physical, cognitive, and emotional challenges. Rehabilitation is crucial for recovery, but outcomes can vary widely. Cognitive reserve (CR) has emerged as a factor influencing these outcomes. This systematic review evaluates the role of CR in post-stroke rehabilitation, examining whether higher CR is associated with better outcomes. **Methods**: A systematic search of PubMed, Google Scholar, Scopus, and Cochrane Library databases was conducted for studies published between 2004 and 2024. Studies examining social-behavior CR proxies (e.g., education, bilingualism) and their impact on post-stroke outcomes were included. Data were analyzed using descriptive statistics. The study quality was assessed using the Methodological Index for NOn-Randomized Studies (MINORS) scale. **Results**: Among 3851 articles screened, 27 met the inclusion criteria. Higher education levels, bilingualism, and engagement in cognitively stimulating activities were associated with better cognitive outcomes and functional recovery. Lower socioeconomic status (SES) correlated with poorer outcomes. Early rehabilitation and dynamic CR proxies showed stronger associations with cognitive recovery than static ones. **Conclusions:** CR may predict post-stroke rehabilitation outcomes, with education, bilingualism, and active engagement in cognitive activities showing potential benefits. Future research should explore CR’s role alongside factors like lesion location and severity in enhancing recovery.

## 1. Introduction

Stroke is one of the leading causes of disability and death worldwide, imposing significant individual and societal burdens [1,2,3]. Survivors often face a spectrum of physical, cognitive, and emotional challenges that can severely impact their quality of life [4,5]. Rehabilitation is critical in mitigating these effects, enhancing recovery and maximizing functional independence [6]. However, outcomes can vary widely among individuals, raising the question of what factors contribute to this variability. One emerging concept that has gained considerable attention in recent years in stroke recovery is that of cognitive reserve (CR) [7]. CR represents the brain’s resilience to cope with neurological damage and is believed to be built over a lifetime of intellectual, social, and physical activities [7,8,9].

The notion of CR was first introduced in the context of Alzheimer’s disease, where it was observed that individuals with higher CR could tolerate greater degrees of brain pathology before exhibiting clinical symptoms [10,11]. This concept has since been extended to other neurological conditions, including multiple sclerosis and stroke [7,12,13]. The underlying hypothesis is that a higher CR could provide a protective buffer, enhancing the brain’s ability to compensate for the damage caused by a stroke and improving the efficacy of rehabilitation interventions [14,15].

CR is believed to be shaped by various factors, such as education, occupation, and lifelong engagement in cognitively stimulating activities [9,10,16]. CR is a multifaceted construct, but there are no standardized measurement guidelines to investigate it, with studies frequently relying on indicators, namely socio-behavioral factors (education, occupation, engagement in leisure activities, intelligence indexes, neuropsychological assessment), electrophysiology (EEG/MEG, event-related potentials, functional connectivity), neuroimaging (PET, fMRI, brain size/volume, atrophies, neural activation, functional connectivity), and genetic proxies [8].

Since different studies use various proxies to estimate CR, the effects observed can differ, leading to inconsistent findings [17]. Understanding how cognitive reserve interacts with post-stroke rehabilitation could provide valuable insights into tailoring more effective and individualized therapeutic approaches [18].

This systematic review aims to summarize the current evidence on the role of cognitive reserve. While the body of evidence exploring electrophysiology, neuroimaging, and genetic proxies is growing, there is abundant literature on social–behavioral metrics. Thus, we specifically focused on the relationships between this metric and rehabilitation outcomes in post-stroke subjects. It will examine whether individuals with higher CR show better functional recovery, greater cognitive improvement, and a higher quality of life following stroke rehabilitation. By combining insights from various studies, this review provides a comprehensive understanding of how CR may influence the rehabilitation process.

## 2. Materials and Methods

### 2.1. Study Design

This study was conducted according to the Preferred Reporting Items for Systematic Reviews and Meta-Analyses (PRISMA) statement [19,20]. The protocol has not been registered on PROSPERO or any other platform.

### 2.2. Eligibility Criteria

This review included studies based on the following criteria: (1) research involving human participants aged 18 years or older who suffered an ischemic or hemorrhagic stroke; (2) cohort studies, either prospective or retrospective; (3) studies examining CR proxies such as education, occupation, intellectual quotient (IQ), bilingualism, leisure activities, and socioeconomic status as the primary variables; (4) those reporting on mortality and functional outcomes, including post-stroke cognitive impairment; and (5) a minimum follow-up period of three months. Exclusions were applied to studies focusing on transient ischemic attack (TIA), studies not involving first-time stroke patients, studies not in English, and non-primary research formats such as conference abstracts, letters, comments, editorials, and case reports. Systematic reviews and meta-analyses were also excluded, though their reference lists were reviewed to find relevant primary studies.

### 2.3. Search Strategy

The search strategy was developed in collaboration with an expert librarian. Four journal search and indexing databases (PubMed, Google Scholar, Scopus, and Cochrane Library) were used to systematically examine the scientific evidence from January 1, 2004, to July 31, 2024. We used the Rayyan (https://new.rayyan.ai/, last accessed on 3 November 2024) to create a database containing the search results. The search focused on these terms: “cognitive reserve,” “stroke,” and “rehabilitation” or “recovery.” Detailed MeSh-term branching is reported in Appendix B.

### 2.4. Study Selection, Data Collection, and Data Extraction

Four authors (D.B., S.B., D.S., and A.D.T.) independently identified each article’s title, abstract, and keywords and evaluated its eligibility. The following data were collected: study characteristics (first author, publication year, country, journal, and study design), demographic data (age and proportion of men/women), population recruitment interval, stroke types, the length of follow-up, CR indicators, outcome definition and assessment, the type of statistical model, main findings, and the relationship between CR and the outcome. Two reviewers (S.B. and D.B.) extracted data from all the included full-text manuscripts using a predefined and standardized data extraction form. The data collected from each article included information on the general description of the study, participants, methodological characteristics, and results. D.S. and A.D.T. independently verified this process, and C.C. resolved disagreements among the four reviewers.

### 2.5. Outcome Definition

The prespecified primary outcome of interest was functional outcomes, including post-stroke cognitive improvement, functional ability, and psychological well-being, including reduced levels of depression or anxiety.

### 2.6. Risk of Bias and Quality Assessment

The quality and level of evidence of each study were assessed independently by four authors (D.B., S.B., D.S., and A.D.T.). The quality of evidence was evaluated based on internal validity factors (such as study design, reporting quality, presence of selection, misclassification biases, and potential confounding) and external validity (generalizability). This evaluation was conducted using the “Methodological Index for NOn-Randomized Studies” (MINORS) quality assessment tool [21], which assigns a score to each study out of a total of 24 points. The MINORS tool is a validated instrument designed to evaluate the methodological quality of non-randomized studies in comparative and non-comparative contexts. It includes 12 criteria, scored from 0 to 2, for a maximum score of 24 for comparative studies. The criteria assess aspects such as the clarity of the study aim, the inclusion of consecutive patients, the prospective collection of data, and the adequacy of follow-up. Each study’s evidence level was categorized based on the Oxford Centre of Evidence-Based Medicine (OCEBM) model [22]. According to the study exclusion criteria, levels 1a, 2a, 3a (systematic reviews), 4 (case series), and 5 (opinion-based papers) were not included. We evaluated the potential bias of each study in terms of study participation, attrition, prognostic factor measurement, outcome measurement, study confounding, and statistical analysis. Discrepancies in MINORS scoring or OCEBM categorization were resolved by the last author (C.C.).

### 2.7. Data Analysis

The analysis aimed to summarize the findings from the included studies and to provide a comprehensive overview of the role of CR in improving functional outcomes in patients who undergo rehabilitation after a stroke. To analyze the data we collected, we used descriptive statistics. We reported relevant data using means, medians, and outcome ranges. All calculations, figures, and tables were created using Microsoft Excel and Word [23].

### 2.8. Data Synthesis

We summarized the extracted data, the quality, and the level of evidence for each study.

## 3. Relevant Sections

An overview of the study identification process is provided in Figure 1.

A detailed breakdown of the search process is summarized in Appendix A. Following the identified MeSH terms and keywords, the initial research yielded 3851 articles. Of these articles, 48 duplicates were removed, leaving 3811 potentially relevant articles. A total of 3760 articles were removed after an initial screening of titles and abstracts to determine if the studies met the inclusion and exclusion criteria, while 51 were selected. The full texts were further narrowed to 27 full manuscripts based on the inclusion/exclusion criteria and included in the systematic review. Included studies were published between 2004–2024. We chose this period even though the first theoretical study on CR was published in 2002 [24]. However, the initial research about the application of CR in the clinical field of stroke was released in 2004 [25], and this year marks the twentieth anniversary since then.

### 3.1. Study Characteristics and Quality

The studies were published between 2004 and 2024, with the majority originating from the European Union (n = 13; 48%), followed by the USA (n = 6; 22%), Asia (n = 3; 11%), Australia (n = 2; 7%), Switzerland (n = 1; 4%), the UK (n = 1; 4%), and Canada (n = 1; 4%). Twenty-two (81%) were retrospective, while four (17%) were prospective in design, with a median sample size of 291 individuals and a range from 10 to 12,561 across studies.

The MINORS scale was used to evaluate the methodological quality of each study. Due to the observational nature of the studies, two items on the MINORS scale only applied to some included studies, preventing total scores. The maximum possible score for the eligible studies was 20. The mean MINORS score was 13.3 (range: 10–19 out of 20); the median was 13. One study (4%) scored 19, one study (4%) scored 18, one study (4%) scored 17, two studies (7%) scored 16, five studies (13.5%) scored 14, seven studies (26%) scored 13, five studies scored 12 (13.5%), and the remaining five studies (22%) scored 10. Most studies scoring at most 17 exhibited limitations, such as incomplete sample representation descriptions. All quality assessment scores are reported in Table 1.

All the reviewed studies demonstrated the highest level of evidence, categorized as level 2b according to the OCEBM Levels of Evidence Working Group [22], signifying individual cohort studies.

### 3.2. Outcome Measures

Seven studies used various cognitive testing scales to examine the correlation between cognitive reserve (CR) proxies and post-stroke cognitive outcomes [26,27,28,29,30,31,32]. Among the CR indicators, education level was most frequently used (37%), consistently showing that higher education is associated with better cognitive outcomes, thus highlighting education as a robust proxy for CR [14,28,33,34,35,36,37,38,39]. Other CR indicators included bilingualism (7%) [35,40], occupational or socioeconomic status (7%) [41,42], engagement in leisure and social activities (11%) [30,43,44], and premorbid IQ (4%) [33]. Two studies (7%) [45,46] used imaging proxies, showing that this metric is associated with favorable outcomes regardless of age or lesion (region or size). Factors related to cognitive impairment, as assessed by the MoCA (4%) [28], the rehabilitation timing, and specific interventions, including early rehabilitation (4%) [29] and attention process training (4%) [47], were also examined. Cognitive impairment was demonstrated to be a significant predictor of cognitive reserve in people with stroke [28]. Early rehabilitation and attention process training were associated with better neuropsychological outcomes and cognitive test performance.

### 3.3. CR and Cognitive and Functional Outcomes After Stroke

Most studies (18/27) included in this review address ischemic stroke. Only five include both ischemic and hemorrhagic cases, and four encompass cerebrovascular, traumatic, or other brain injuries, as depicted in Figure 2.

In the few studies that do include hemorrhagic stroke patients [14,43], they are grouped with ischemic stroke patients and typically represent a minority of the sample.

Furthermore, the studies included in this systematic review focused on socio-behavioral proxies [7], such as education, bilingualism, occupation, leisure activities, premorbid intellectual quotient (IQ), and socioeconomic status (SES) [41]. Longitudinal studies indicate that education significantly influences cognitive recovery after a stroke. Higher education levels are associated with faster cognitive recovery within the first three months post-stroke [14], and it has been shown that education may moderate the impact of stroke on cognitive impairment [13], even though different studies have reached different conclusions [13]. For instance, Gil-Pagés et al. [32] did not find significant associations between the number of years of education and cognitive changes (e.g., attention, memory, executive functions) in the chronic stage of stroke [13]. Noteworthily, Umarova et al. [48] showed that higher education in combination with a younger age could mitigate cognitive decline when in the presence of clinical conditions characterized by larger lesion sizes. In comparison, lower education and older age increased the probability of worsening cognitive functioning, even if the lesions were small [48]. Abdullah et al. [28] also highlighted education as a stronger predictor of cognitive dysfunction than age or sex in the chronic stage of stroke. In acute stroke patients, more years of education have been linked to better cognitive performance in specific domains, as well as in overall cognition [49]. Additionally, patients with a higher level of education (>12 years) who had a stroke within the past 24 h exhibited better linguistic performance [37].

Bilingualism was shown to have a protective effect against post-stroke cognitive impairment. Bilingual patients exhibited better cognitive performance and faster recovery on cognitive tasks compared with monolinguals [35,40]. Engagement in leisure activities and social ties significantly predict slower cognitive decline and better cognitive resilience post-stroke [30,43,44]. These authors also showed that physical activity and social participation were linked to higher cognitive performance, suggesting their role in enhancing CR and promoting better cognitive health post-stroke [27,30,31,43,44]. Noteworthily, these activities were found to contribute to the building of cognitive reserve over a lifetime. It has been shown that the timing of rehabilitation interventions plays a crucial role in cognitive recovery [29]. Premorbid IQ, as measured by the National Adult Reading Test (NART), was a strong predictor of cognitive outcomes 1-year post-stroke, more so than stroke severity or vascular risk factors. Makin et al. [33] used premorbid IQ and education as CR indicators, demonstrating that they are good predictors of cognitive recovery. It has also been shown that individuals with a higher premorbid IQ tend to experience better outcomes after a stroke, even when IQ is assessed through the Mini-Mental State Examination (MMSE), Activities of Daily Living (ADL), and Instrumental Activities of Daily Living (IADL) scores [33,50]. Additionally, cognitive impairment assessed through the Montreal Cognitive Assessment (MoCA) appears to be a good predictor of functional outcomes after a stroke.

Additionally, using the modified Rankin Scale (mRS), some studies found a strong connection between CR and post-stroke disability [28,45,46,49,51]. Some authors used socioeconomic status (SES) as a CR indicator, even though the definitions of this measure, which includes education, occupation, income [14,52,53,54], medical insurance, and neighborhood status [54], vary across the studies [14,42,52,53], marital status, place of residence [55], income, caregiver presence, and insurance [53] have been used to define socioeconomic status as well. Nonetheless, lower SES is consistently associated with a higher risk of adverse clinical outcomes in ischemic stroke patients, in line with what was reported by Tao et al. [56]. The relationship between SES and motor impairment post-stroke has been explored using the Barthel Index (BI) [57]. That study provides supportive evidence that lower educational attainment is linked to poorer functional outcomes [57].

Furthermore, socioeconomic disadvantages indicated by patient postcodes are significantly associated with both short- and long-term motor impairments following a stroke [52,53,54]. Similarly, Shin et al. [14] found that individuals in lower occupational positions exhibited poorer cognitive performance and had a higher risk of cognitive impairment immediately after a stroke and 30 months later. Altogether, despite a few inconsistencies, it is demonstrated that higher occupation and education levels are linked to better cognitive recovery within three months after a stroke [14,58].

### 3.4. Rehabilitation, Occupation, and Cognitive Stimulating Activities

Many studies have examined the impact of occupation and involvement in intellectual activities on cognitive outcomes following a stroke. Gil-Pagés et al. [32] distinguished objective/static proxies (e.g., education and occupation) and subjective/dynamic proxies (e.g., activities of daily living, hobbies, and social interactions). The study found that static indicators were not linked to cognitive performance in the chronic stage of stroke. At the same time, there was a positive relationship between dynamic indicators and self-perceived metacognitive, attentional, and functional abilities [32]. Ihle et al. [30] supported these findings, reporting that stroke patients who regularly engaged in cognitively stimulating activities such as reading, visiting museums, and traveling performed better on the Trail Making Test (TMT) [13]. Notably, factors like stroke history, gender, and other CR proxies such as education or occupation did not significantly predict changes in TMT scores.

Social integration is also crucial for stroke recovery. According to Glymour et al. [43], stroke patients with strong interpersonal relationships and greater emotional support showed better scores on cognitive tests measuring attention, language, memory, and executive functions six months after the stroke. Although social relationships did not affect cognitive progress in the long term, emotional support was significantly associated with cognitive improvements [43].

## 4. Discussion

The adult brain’s capacity to cope after a brain injury is well documented, as is the property of the brain to modify itself based on experience and aging [59]. Factors in an adult’s environment and health can influence the brain’s flexibility to adapt and maintain cognitive skills or to cope with pathology over the lifespan [10,60]. While theoretical models continue to evolve, cognitive reserve generally refers to an individual’s ability to sustain adequate cognitive performance despite neurodegeneration [24,61,62]. Given that stroke is a highly debilitating condition with direct impacts on individuals, indirect effects on families, and societal costs [63], understanding an individual’s capacity or potential for recovery is a challenge in the field of stroke rehabilitation when trying to improve the quality of life of these patients and their caregivers and to reduce the global burden [7]. It is well known that factors specific to stroke, along with demographic variables like infarct location, extent of brain damage, clinical complications, age, gender, and vascular risk factors, contribute significantly to the variability observed in treatment responses and recovery outcomes among individuals [1]. However, increasing evidence suggests that premorbid factors, such as cognitive reserve, may also contribute to the variability of therapeutic and rehabilitation responses [7]. The systematic review aimed to assess the role of cognitive reserve (CR), measured by socio-behavioral proxies, in post-stroke rehabilitation outcomes, revealing significant insights into several variables of this metric and their associations with cognitive and functional recovery. Hemorrhagic stroke is less frequently studied due to its lower prevalence, higher mortality rates, and the complexities involved in its clinical management. As a result, the existing evidence predominantly focuses on ischemic stroke, underscoring the need for further studies to explore the role of CR in the rehabilitation of hemorrhagic stroke patients. Overall, the studies included in this systematic review highlight the complexity of cognitive reserve and its substantial influence on rehabilitation outcomes after a stroke. Factors such as education, bilingualism, and social activities are all associated with improved cognitive recovery and long-term results.

The review highlighted education as a predominant CR proxy, consistently showing that higher educational levels are linked to better cognitive outcomes post-stroke. This finding aligns with previous studies indicating that higher education levels are inversely proportional to the onset of dementia [34,64,65,66]. Education helps with acute cognitive recovery [14] and enhances cognitive performance in chronic stages of stroke, especially when combined with a younger age [48,49]

Other proxies, such as bilingualism, demonstrated a protective effect against cognitive impairment, with bilingual patients showing better cognitive performance and faster recovery than monolinguals [35,67]. Musical abilities also correlated with reduced severity of post-stroke aphasic deficits [68]. Engagement in cognitively stimulating activities and higher occupational levels were positively correlated with better cognitive performance post-stroke [13]. Gil-Pagés et al. [32] distinguished between static (education, occupation) and dynamic (activities of daily living, hobbies, social interactions) CR proxies, with dynamic indicators showing stronger associations with metacognitive, attentional, and functional skills. Moreover, dynamic proxies were positively associated with cognitive functioning in the chronic stage of stroke [32,43], while static proxies (education and occupation) were beneficial for cognitive recovery three months post-stroke [14]. Ihle et al. [30] corroborated these findings, noting that participation in activities like reading, visiting museums, and traveling improved performance on cognitive tests such as the Trail Making Test (TMT).

Socioeconomic status (SES), which includes factors such as education, occupation, income, medical insurance, and neighborhood status, was a crucial determinant of post-stroke outcomes. Studies consistently found that lower SES increased the risk of negative clinical outcomes, emphasizing the multifaceted nature of SES in influencing recovery trajectories. The Barthel Index (BI) further supported the relationship between SES and motor impairment, suggesting that socioeconomic factors should be integrated into rehabilitation strategies to optimize recovery.

Overall, this systematic review shows that socio-behavioral proxies can significantly predict cognitive outcomes in the stroke population. However, further evidence through meta-analyses is needed to support this relationship. However, it is worth noting that such assessments may only partially capture an individual’s overall cognitive engagement and complexity of CR [7]. Fortunately, we have access to advanced techniques such as electrophysiological measures, neuroimaging, and genetic testing, which offer opportunities for a more comprehensive evaluation of CR. Thus, more studies are needed to investigate the relationships between these proxies, CR, and functional outcomes after a stroke.

## 5. Limitations and Future Directions

The studies reviewed often carried limitations such as publication bias toward positive results and insufficient details for meaningful comparisons, particularly regarding rehabilitation treatments. The practice of customizing interventions to individual patient’s needs, while crucial, hinders the use of experimental and standard protocols and, in turn, the comparability across studies. Therefore, future research should ensure that recruitment excludes spontaneous recovery by starting at least six months post-stroke. It should also use large, representative samples, ideally in multi-center studies, and provide detailed information on the type and extent of vascular damage. It would be beneficial to stratify samples based on the years of education and to incorporate long-term follow-up periods. It would also be crucial to utilize different representative CR proxies and to specify rehabilitation protocols. Crucially, integrating advanced assessment methods such as electrophysiological techniques, neuroimaging, and genetic testing can significantly enrich the evaluation of CR [7]. Utilizing a diverse range of CR proxies alongside clearly specified rehabilitation protocols will improve the quality and reliability of future studies. Finally, employing multiple or the most representative cognitive and functional outcome scales will contribute to a more nuanced understanding of the interplay between CR and rehabilitation outcomes.

## 6. Conclusions

This review underscores the importance of considering CR in post-stroke rehabilitation to improve patient recovery. While the current literature primarily addresses CR’s protective or predictive role in cognitive outcomes and cognitive decline, it often lacks comprehensive details on rehabilitation treatments. This leaves substantial gaps in understanding the modulatory effects of CR proxies on rehabilitation protocols and long-term outcomes, making it challenging to draw definitive conclusions.

Recognizing CR as a crucial piece in neuropsychological assessments could provide valuable insights, allowing for a more multidimensional patient evaluation. Specifically, CR may play a role in improving current assessments that help understand and predict recovery trajectories after complex conditions such as stroke [7]. By integrating simple measures of CR alongside stroke severity evaluations, clinicians can obtain a more detailed understanding of prognosis and recovery, including insights into neuropsychological and motor functions and overall quality of life [7]. This, in turn, would facilitate the development of tailored rehabilitation goals that better align with individual patients’ needs. Additionally, to optimize post-stroke recovery and to fully realize the potential of CR, it would be helpful to advance research methodologies and to incorporate individualized rehabilitation strategies, especially in an era marked by significant advancements in digital technologies that enable computerized rehabilitative interventions [58,69].

Finally, acknowledging the importance of CR highlights the necessity for investment in health and preventive medicine: enhancing cognitive reserve through lifestyle choices could positively impact recovery trajectories.

## Figures and Tables

**Figure 1 brainsci-14-01144-f001:**
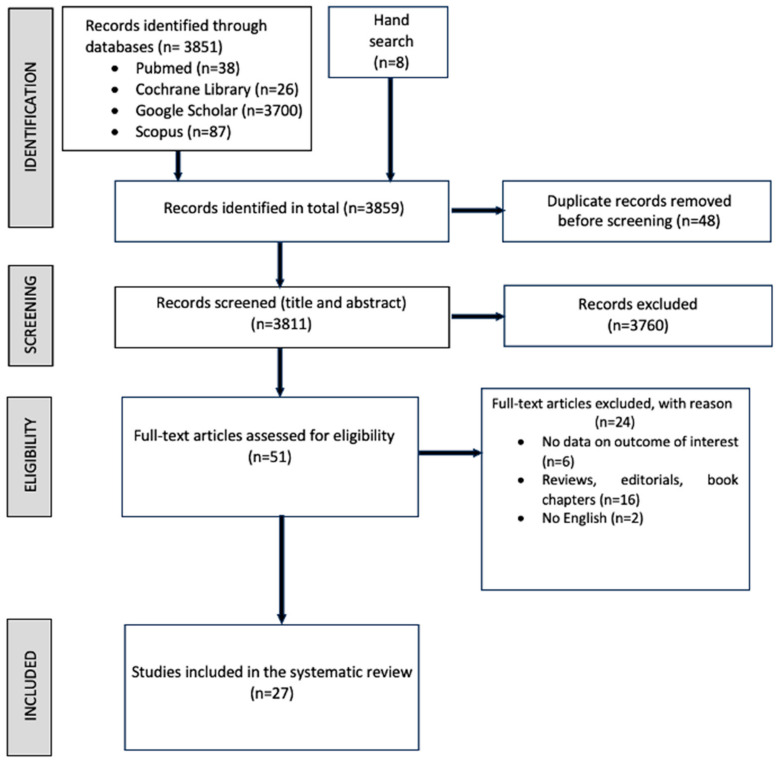
Flow diagram of the literature selection process.

**Figure 2 brainsci-14-01144-f002:**
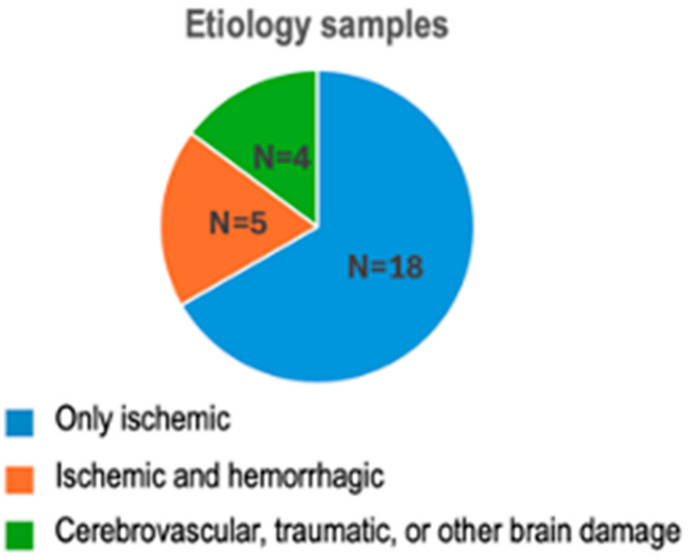
Representation of injury types addressed in included studies.

**Table 1 brainsci-14-01144-t001:** Modified MINORS Scale scores for each study; range: 0–20.

Study Name	A Clearly Stated Aim:	Inclusion of Consecutive Patients:	Prospective Collection of Data:	Endpoints Appropriate to the Aim of the Study	Unbiased Assessment of the Study Endpoint:	Follow-Up Period Appropriate to the Aim of the Study	Loss to f/up Less than 5%	Adequate Group of Control	Contemporary Group	Adequate Statistical Analysis	Score
Adhullah et al., 2021	2	2	2	2	2	0	0	0	0	0	10
Alladi et al., 2015	2	2	2	2	2	2	0	0	0	0	12
Bartfai et al., 2022	2	2	2	2	2	0	0	2	2	2	16
Basagni et al., 2023	2	2	1	2	2	2	2	0	0	0	13
Bertoni et al., 2020	2	2	2	2	2	0	0	0	0	0	10
Dekhtyar et al., 2020	2	2	2	2	2	0	0	0	0	0	10
Durrani et al., 2021	2	2	2	2	2	0	0	0	0	0	10
Elkins et al., 2006	2	2	2	2	2	2	1	0	0	0	13
Gil Pagés et al., 2019	2	2	2	2	2	2	0	0	0	0	12
Glymour et al., 2008	2	2	2	2	2	2	2	0	0	0	14
Gonzalez-Fernandez et al., 2011	2	2	2	2	2	0	0	2	2	2	16
Ihle et al., 2020	2	2	2	2	2	2	0	0	0	0	12
Makin et al., 2018	2	2	2	2	2	2	1	0	0	0	13
Mirza et al., 2016	2	2	2	2	2	2	1	0	0	0	13
Ojala-Oksala et al., 2012	2	2	2	2	2	2	1	0	0	0	13
Padua et al., 2020	2	2	2	2	2	2	0	2	2	2	18
Rosenich et al., 2022	2	2	2	2	2	2	1	0	0	0	13
Schirmer et al., 2019	2	2	2	2	2	2	0	0	0	0	12
Shin et al., 2020	2	2	2	2	2	2	1	0	0	0	13
Skoog et al., 2017	2	2	2	2	2	2	0	2	1	2	17
Umarova et al., 2021	2	2	2	2	2	2	2	0	0	0	14
Umarova et al., 2019	2	2	2	2	2	0	0	0	0	0	10
Withall et al., 2021	2	2	2	2	2	2	1	2	2	2	19
Elayoubi et al., 2022	2	2	2	2	2	2	0	0	0	2	14
Rojas Albert et al., 2022	2	2	2	2	2	2	0	0	0	2	14
Sadeghihassanabadi et al., 2022	2	2	2	2	2	2	0	0	0	2	14
Bettger et al., 2014	2	2	2	2	2	2	0	0	0	1	12

## Data Availability

No new data were created or analyzed in this study. Data sharing is not applicable to this article.

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
