# Peer review of "The Role of Cognitive Reserve in Post-Stroke Rehabilitation Outcomes: A Systematic Review"

_brainsci, 2024, doi:10.3390/brainsci14111144_

Round 1
Reviewer 1 Report
Comments and Suggestions for Authors
The manuscript includes a very interesting item. However, I think the information provided in it is very confusing. Therefore, I have most concerns about the convenience of accepting the current manuscript for publication in this form. Some concrete comments would be as follows:
The manuscript has a high level of similarity to existing published work. It is crucial to reduce plagiarism by properly paraphrasing the text, citing sources appropriately, and ensuring originality. I strongly recommend reviewing the manuscript to eliminate any copied or overly similar text.
Include in the supplementary materials the Preferred Reporting Items for Systematic Reviews and Meta-Analyses (PRISMA) statement (complete).
The protocol of systematic review is not registered in the PROSPERO platform.
Avoid Redundant Abbreviations: In the manuscript, certain terms such as "cognitive reserve (CR)" are defined with abbreviations but are not used again throughout the text.
The methods section is confusing. I do not understand why the list of acronyms is important here. (please write in the appendix or in supplementary materials.
The authors should write descriptors of the above tables and below figures.
Table S2. Characteristics of the included studies. Is the position in the manuscript or in supplementary materials? Please standardize the information in the table.
Results seem Discussion: The Results section currently reads more like a Discussion. It includes speculative statements and interpretations that should be reserved for the Discussion section. I recommend revising this section to focus strictly on presenting the data and factual findings without interpretation or comparison with the existing literature. You could create a table divided by stroke type (ischemic or hemorrhagic) or outcome measures for more clearly.
Author Response
Reviewer 1
The manuscript includes a very interesting item. However, I think the information provided in it is very confusing. Therefore, I have most concerns about the convenience of accepting the current manuscript for publication in this form.
R. Thanks for their thorough and constructive feedback on our manuscript. We appreciate their insights and suggestions to enhance the quality and clarity of our work. Below, we address each of their comments in detail.
Some concrete comments would be as follows:
The manuscript has a high level of similarity to existing published work. It is crucial to reduce plagiarism by properly paraphrasing the text, citing sources appropriately, and ensuring originality. I strongly recommend reviewing the manuscript to eliminate any copied or overly similar text.
R. We acknowledge their concerns regarding the similarity to existing published work. We conducted a comprehensive review to rephrase similar sentences. We also verified all citations to ensure appropriate acknowledgment of prior work, and we used tools such as Grammarly to confirm originality before resubmission. The rewritten sentence has been highlighted in the main text:
Results, lines: 225-229 (p.18); 232-236 (p.18-19); 239-240 (p. 19); 255 (p.19); 263-268 (p. 19); 276-277 (p. 19)
Include in the supplementary materials the Preferred Reporting Items for Systematic Reviews and Meta-Analyses (PRISMA) statement (complete).
R. We have included the full PRISMA checklist, previously submitted in the cover letter, as supplementary materials.
The systematic review protocol is not registered on the PROSPERO platform.
R. We explicitly stated that the protocol has not been registered on the PROSPERO or any other platform.
Material and Methods (lines 78-80; p. 2):
“This study was conducted according to the Preferred Reporting Items for Systematic Reviews and Meta-Analyses (PRISMA) statement [19,20]. The protocol has not been registered on the PROSPERO or any other platform.”
Avoid Redundant Abbreviations: In the manuscript, certain terms such as "cognitive reserve (CR)" are defined with abbreviations but are not used again throughout the text.
R. Thanks for this observation. We revised the manuscript to ensure that abbreviations like "cognitive reserve (CR)" are only defined if used multiple times. Unnecessary abbreviations have been removed to enhance readability.
The methods section is confusing. I do not understand why the list of acronyms is important here. (please write in the appendix or in supplementary materials.
R. We know now that including the list of acronyms within the main Methods section may be distracting. We moved this list to the appendix B, as they suggested. The text has been modified as follows (line 497, p. 24):
“Appendix B. List of acronyms.”
The authors should write descriptors of the above tables and below figures.
R. We added descriptive captions above the tables and below the figures, detailing their contents to enhance clarity and comprehension.
Table S2. Characteristics of the included studies. Is the position in the manuscript or in supplementary materials? Please standardize the information in the table.
R. Table S2 is placed within the supplementary materials: now it appears in the main text, but it will be moved eventually. We renamed it S1, and we standardized its format for uniformity and ease of reference. The text of the caption has been modified as follows (p.15):
Table S1. Characteristics of the included studies. Legend: ACE-R: Addenbrooke’s Cognitive Examination-Revised; ADL/BADL: Activity Daily Living; APT: Attention Process training; ARAT: Action Research Arm Test; BAA: Bilingual Aphasia Adults; BAT: Bilingual Aphasia Test; BDI: Beck Depres-sion Inventory; BHA: Bilingual Healthy Adults; BI: Barthel index; BNT: Boston Naming test; CDR: Clinical Dementia Rating scale; CIQ: Community Integration Questionnaire; CRS Cognitive Reserve Scale; CNS: Central Nervous System; CRIq: Cogni-tive Reserve Index; DoC: Disorder of consciousness; DRS: Disability Rating Scale; DSM: Diagnostic and Statistical Manual of Mental Disorders; DSST: Digit Symbol Substitution Test; DWI: diffusion weighted imaging; EEG: Electroencephalography; GCS: Glasgow Coma Scale; GOS-E: Glasgow Outcome Scale- Extended; HADS: Hospital Anxiety and Depression Scale; K-MMSE: Korean mini-mental state examination; LCF: Level of Cognitive Functioning; LDST: letter-digit substitution task; LUQ: Language Use Questionnaire; mBI: modified Barthel index; MAA: Monolingual Aphasia Adults; MCA: middle cerebral artery; M- DASS-21:Malaysian Depression and Anxiety Screening Scales 21; MCI: Mild Cognitive Impairment; MHA: Monolingual Healthy Adults; MoCA: Montreal Cognitive Assessment; Modified MMSE: 3MS Modified Mini mental State Examination; MRI: Magnetic Resonance Imaging; mRS: modified Rankin Scale; NART: National Adult Reading Test; NIHSS: National Institute of Health Stroke Scale; NLTT: Non-linguistic Triad task; PCRS: Patient Competency Rating Scale; PMR test: Spanish version of FAS test- phonemic verbal fluency; RBMT: River mead Behavioural Memory Test; SES: Socio-economic status; Tele-GEMS: Tele-Global Examination of Mental State; TBI: Traumatic Brain Injury; TIA: Transient Ischemic Attack; TMT: trail Making Test; VLV: Vivre-Leben-Vivere; WAIS: Wechsler Adult Intelligence Scale; T2-FLAIR: T2 fluid- attenuated inversion recovery; VFT: verbal fluency test; WHOQOL BREF-2: World Health Organization-Quality of Life- Bref-2.; WMH: White Matter Hyperintense-ties; 15-WLT:15-word verbal learning test.
Results seem Discussion: The Results section currently reads more like a Discussion. It includes speculative statements and interpretations that should be reserved for the Discussion section. I recommend revising this section to focus strictly on presenting the data and factual findings without interpretation or comparison with the existing literature. You could create a table divided by stroke type (ischemic or hemorrhagic) or outcome measures for more clearly.
R. We appreciate their observation regarding the overlap between the Results and Discussion sections. We revised the Results section to present the findings as factual data, reserving all speculative or interpretative content for the Discussion section. Additionally, we have created one graph by adding a caption (Figure 2. Representation of injury types addressed in included studies, p. 18) to show the distribution of injury types considered in the studies we included in this review. We also tried to verify if there were significant differences in mean quality methodology among these studies, plotting MINORS score by stroke type (only ischemic, ischemic hemorrhagic, and cerebrovascular, traumatic, or other brain damage; see below), but there were no differences, and given the low number of studies, we did not include it in the review.

Reviewer 2 Report
Comments and Suggestions for Authors
Review report for “The Role of Cognitive Reserve in Post-Stroke Rehabilitation 2 Outcomes: A Systematic Review”
Comment to the authors
In the present systematic review, the authors investigated the studies on the cognitive reserve in rehabilitation. While it is widely recognized by clinicians that cognitive function is key to the implementation of qualified rehabilitative treatments, researchers have revealed that cognitive reserve is also important in recovery after stroke. The authors investigated 99 articles and systematically reviewed the 23 studies, and found that higher education levels, bilingualism, and engagement in cognitively stimulating activities were associated with better cognitive outcomes and functional recovery, while lower socioeconomic status correlated with poorer outcomes. In addition, the strength of this systematic review is that it reveals the treatment aspect, namely, early rehabilitation and dynamic cognitive reserve proxies, which showed stronger associations with cognitive recovery than static ones.
The most significant weak point of this systematic review is, as the authors described in the Introduction section, “Since cognitive reserve cannot be directly measured, researchers often rely on proxies like education level, intellectual quotient, and age to estimate it (lines 52-53)”, the definition of cognitive reserve is not extended from this initial definition. Therefore, the main outcome of this systematic review seems still to be caught in this dogma: “education, bilingualism, and active engagement in cognitive activities showing potential benefits” (abstract). This fundamental restriction will limit interest in this issue. The reviewer would expect the authors to propose some implications for further measurement or battery, which may be suited to evaluate cognitive reserve.
Minor concerns;
Another shortage is at term “synaptic plasticity” or “neuroplasticity”. (Lines 54, 270) Although the authors use this term as a key to explain the cognitive reserve of patients, the rationale is not shown. Some evidences from the field of preclinical or basic research field should be presented, otherwise this term should be withdrawn to avoid misunderstanding of the readers.
Line 294, “early rehabilitation intervention”, is not related to the cognitive reserve.
Author Response
Reviewer 2
In the present systematic review, the authors investigated the studies on the cognitive reserve in rehabilitation. While it is widely recognized by clinicians that cognitive function is key to the implementation of qualified rehabilitative treatments, researchers have revealed that cognitive reserve is also important in recovery after stroke. The authors investigated 99 articles and systematically reviewed the 23 studies, and found that higher education levels, bilingualism, and engagement in cognitively stimulating activities were associated with better cognitive outcomes and functional recovery, while lower socioeconomic status correlated with poorer outcomes. In addition, the strength of this systematic review is that it reveals the treatment aspect, namely, early rehabilitation and dynamic cognitive reserve proxies, which showed stronger associations with cognitive recovery than static ones.
The most significant weak point of this systematic review is, as the authors described in the Introduction section, “Since cognitive reserve cannot be directly measured, researchers often rely on proxies like education level, intellectual quotient, and age to estimate it (lines 52-53)”, the definition of cognitive reserve is not extended from this initial definition. Therefore, the main outcome of this systematic review seems still to be caught in this dogma: “education, bilingualism, and active engagement in cognitive activities showing potential benefits” (abstract). This fundamental restriction will limit interest in this issue. The reviewer would expect the authors to propose some implications for further measurement or battery, which may be suited to evaluate cognitive reserve.
R. Thanks for their thoughtful comments and for recognizing the strengths of our systematic review. We appreciate their insights, particularly regarding the limitations of defining and measuring cognitive reserve (CR). In response to their suggestions—and similar feedback from Reviewer 3—we have revised the manuscript to broaden our conceptualization of CR and address alternative ways to measure it.
The main text has been modified to:
-
Extending the definition of Cognitive Reserve: we agree that relying solely on traditional proxies like education level, IQ, and age may restrict the interpretation of cognitive reserve. To address this, we have updated the Introduction and Discussion sections with a more comprehensive definition of CR, incorporating recent advancements and highlighting distinctions among different types of proxies: socio-behavioral, electrophysiology, neuroimaging, and genetic. While our review focused on socio-behavioral proxies (education, occupation, leisure activities, Intelligence Indexes, neuropsychological assessment) due to their prevalence in existing literature, we also acknowledge the growth of studies investigating other types of CR proxies, such as biomarkers and neuroimaging data.
-
Implications for new measures of Cognitive Reserve: based on their suggestion, we expanded our discussion to include alternative tools and methods for assessing cognitive reserve. Specifically, we now discuss the potential of neuroimaging techniques (e.g., fMRI, PET scans), composite measures that integrate lifestyle and physiological data, and neuropsychological batteries focused on cognitive flexibility and resilience. These methods may provide a richer, multidimensional assessment of CR, potentially enhancing predictions of cognitive outcomes and rehabilitation efficacy.
-
Future directions and research recommendations: we strengthened our conclusions by recommending specific areas for future research. We emphasized the need for studies that validate these alternative CR measures in clinical populations, especially within stroke rehabilitation, and suggest longitudinal research on the interaction between dynamic CR proxies and static proxies in influencing recovery outcomes.
The text has been modified as follows:
Introduction (lines: 51-58, 68-71; p. 1-2):
“CR is a multifaceted construct, but there are no standardized measurement guidelines to investigate it, with studies frequently relying on indicators, namely socio-behavioral (education, occupation, engagement in leisure activities, intelligence indexes, neuropsychological assessment), electrophysiology (EEG/MEG, event-related potentials, functional connectivity), neuroimaging (PET, fMRI, brain size/volume; atrophies; neural activation; functional connectivity, and genetic proxies [8].”
And:
“Even though there is a growing body of evidence exploring electrophysiology, neuroimaging, and genetic proxies, there is a great literature on social-behavioral metrics. Thus, we specifically focused on the relationships between this metric and rehabilitation outcomes in post-stroke subjects.”
And Discussion (lines 366-374, p. 21):
“Overall, this systematic review shows that socio-behavioral proxies can significantly predict cognitive outcomes in the stroke population. However, further evidence through meta-analyses is needed to support this relationship. However, it is worth noting that such assessments may only partially capture an individual's overall cognitive engagement and complexity of CR [7]. Fortunately, we have access to advanced techniques such as electrophysiological measures, neuroimaging, and genetic testing, which offer opportunities for a more comprehensive evaluation of CR. Thus, more studies are needed to investigate the relationships between these proxies, CR, and functional outcomes after a stroke.”
And (387-393, p. 20-21):
“Crucially, integrating advanced assessment methods such as electrophysiological techniques, neuroimaging, and genetic testing can significantly enrich the evaluation of CR [7]. Utilizing a diverse range of CR proxies alongside clearly specified rehabilitation protocols will improve the quality and reliability of future studies. Finally, employing multiple or the most representative cognitive and functional outcome scales will contribute to a more nuanced understanding of the interplay between CR and rehabilitation outcomes.”
Minor concerns;
Another shortage is at term “synaptic plasticity” or “neuroplasticity”. (Lines 54, 270) Although the authors use this term as a key to explain the cognitive reserve of patients, the rationale is not shown. Some evidences from the field of preclinical or basic research field should be presented, otherwise this term should be withdrawn to avoid misunderstanding of the readers.
R. Thanks for pointing out the need for further clarification on "synaptic plasticity" or "neuroplasticity" related to cognitive reserve in stroke rehabilitation. We understand that in this shape, this term can cause misunderstanding among the readers, and for this reason, we withdraw them. The text has been modified in this way:
Introduction
“CR is believed to be shaped by various factors, such as education, occupation, and lifelong engagement in cognitively stimulating activities [9, 10,16]. CR is a multifaceted construct, but there are no standardized measurement guidelines to investigate it, with studies frequently relying on indicators, namely socio-behavioral (education, occupation, engagement in leisure activities, intelligence indexes, neuropsychological assessment), electrophysiology (EEG/MEG, event-related potentials, functional connectivity), neuroimaging (PET, fMRI, brain size/volume; atrophies; neural activation; functional connectivity, and genetic proxies [8].”
And
Discussion (lines 311-313; p.20):
“The adult brain's capacity to cope after a brain injury is well documented, as well as the property of the brain to modify itself based on experience and aging [56].”
Line 294, “early rehabilitation intervention”, is not related to the cognitive reserve.
R. It has been removed now. The sentence reads as follows (Discussion, lines 336-338; p.20):
“Factors such as education, bilingualism, and social activities are all associated with improved cognitive recovery and long-term results.”

Reviewer 3 Report
Comments and Suggestions for Authors
This manuscript aims at providing a review of the studies investigating the role of cognitive reserve (CR) in rehabilitation prognosis after stroke. This topic is indeed, very important in the field of stroke prevention and rehabilitation. Being clearly written, nevertheless, the manuscript is not acceptable in the present form since, in my opinion, the authors have significantly reduced the search of the relevant publications by making an unjustified assumption that “cognitive reserve cannot be directly measured” (p.2) and limiting their search only to such proxies as “education, occupation, intellectual quotient (IQ), bilingualism, leisure activities, and socioeconomic status as the primary variables” (p.2). At the same time a large body of studies (Stiekema et al., 2024; Salvadori et al. 2022; Kulesh et al., 2019; Chiti et al., 2014) highlight the potential of the Montreal cognitive assessment scale to predict stroke outcomes, both at discharge and in a long term. And thus, this metrics should be also included in the review.
Furthermore, I have a strong concern about the selection of the terms (‘cognitive reserve,’ ‘stroke,’ and ‘rehabilitation.’) used for the search across databases. If you replace ‘rehabilitation’ with ‘recovery’, this will lead to a much larger number of potentially relevant publications. Indeed, in a recent review “The role of cognitive reserve in ischemic stroke prognosis: A systematic review” by Tao et al (2023) have identified (between 2004 and 2022) 2526 (!) potentially relevant studies, from which they have selected 28 studies, limiting their search to ischemic stroke. On the contrary, in this manuscript the search period is larger (2004 – 2024), the type of stroke is not limited to the ischemic type but the authors have included only 23 studies! This makes me very doubtful about the search strategy and the contribution of this review to the field.
Author Response
Reviewer 3
This manuscript aims at providing a review of the studies investigating the role of cognitive reserve (CR) in recovery prognosis after stroke. This topic is indeed, very important in the field of stroke prevention and recovery. Being clearly written, nevertheless, the manuscript is not acceptable in the present form since, in my opinion, the authors have significantly reduced the search of the relevant publications by making an unjustified assumption that “cognitive reserve cannot be directly measured” (p.2) and limiting their search only to such proxies as “education, occupation, intellectual quotient (IQ), bilingualism, leisure activities, and socioeconomic status as the primary variables” (p.2).
R. Thank you for your thoughtful and constructive feedback. We appreciate your insights, which have helped us to consider ways to strengthen the manuscript. We recognize the limitations of focusing solely on proxies like education, occupation, and IQ, and we understand your concern regarding the assumption that CR cannot be directly measured. As we replied to reviewer 2, to address this issue, we have rephrased the introduction to better convey the complexities involved in defining and measuring CR. We now explain that CR is a multifaceted construct without standardized measurement guidelines, with studies frequently relying on indicators like educational level, occupational complexity, and engagement in leisure activities. These proxies, while informative, may not fully reflect CR at an individual level, as supported by.Pappalettera et al., 2024. We have made clearer distinctions between socio-behavioral, electrophysiology, neuroimaging, and genetic proxies, thus offering a comprehensive overview of the metrics commonly employed in CR research.
At the same time a large body of studies (Stiekema et al., 2024; Salvadori et al. 2022; Kulesh et al., 2019; Chiti et al., 2014) highlight the potential of the Montreal cognitive assessment scale to predict stroke outcomes, both at discharge and in a long term. And thus, this metrics should be also included in the review.
R. Thanks for pointing this out. We followed their suggestion and included the cognitive impairment, as assessed by MoCA, in the Results of this review. The text was amended in this way:
Results (lines 194-198; p.18):
“Factors related to cognitive impairment, as assessed by the MoCA (4%) [28], the rehabilitation timing and specific interventions were also examined, with early rehabilitation (4%) [29] and attention process training (4%) [42]. Cognitive impairment was demonstrated to be a significant predictor of cognitive reserve in people with stroke [28]. Early rehabilitation and attention process training were associated with better neuropsychological outcomes and cognitive test performance.”
And (lines 266-268; p. 19):
“Additionally, cognitive impairment assessed through the Montreal Cognitive Assessment (MoCA) appears to be a good predictor of functional outcomes after a stroke.”
Furthermore, I have a strong concern about the selection of the terms (‘cognitive reserve,’ ‘stroke,’ and ‘recovery.’) used for the search across databases. If you replace ‘recovery’ with ‘recovery’, this will lead to a much larger number of potentially relevant publications. Indeed, in a recent review “The role of cognitive reserve in ischemic stroke prognosis: A systematic review” by Tao et al (2023) have identified (between 2004 and 2022) 2526 (!) potentially relevant studies, from which they have selected 28 studies, limiting their search to ischemic stroke. On the contrary, in this manuscript the search period is larger (2004 – 2024), the type of stroke is not limited to the ischemic type but the authors have included only 23 studies! This makes me very doubtful about the search strategy and the contribution of this review to the field.
R. We understand the concern regarding our current selection of 23 studies over 20 years, as compared to other reviews with similar criteria, such as Tao et al. (2023), which identified 28 studies on ischemic stroke within a shorter timeframe. To address this, we modified our search strategy to confirm that all relevant studies are incorporated, ensuring comprehensive coverage and a clear contribution to the field.
The new search yielded 3851 records and new the search strategy is as follows (lines 449-494, p.23):
(("Rehabilitation Centers"[MeSH Terms] OR "Rehabilitation Research"[MeSH Terms] OR "Rehabilitation"[MeSH Terms] OR "Rehabilitation"[MeSH Subheading] OR "Physical and Rehabilitation Medicine"[MeSH Terms] OR "hospitals, rehabilitation"[MeSH Terms] OR "Rehabilitation"[Title/Abstract] OR "Recovery Centers"[MeSH Terms] OR "Recovery Research"[MeSH Terms] OR "Recovery"[MeSH Terms] OR "Recovery"[MeSH Subheading] OR "Physical and Recovery Medicine"[MeSH Terms] OR "hospitals, recovery"[MeSH Terms] OR "Recovery"[Title/Abstract] OR "recoverys"[Title/Abstract]
OR "rehabilitations"[Title/Abstract] OR "Exercise Therapy"[MeSH Terms] OR "Exercise"[Title/Abstract] OR "exercises"[Title/Abstract] OR "Physical and Rehabilitation Medicine"[MeSH Terms] OR "reeducation"[Title/Abstract] OR "reeducations"[Title/Abstract] OR "training"[Title/Abstract] OR "trainings"[Title/Abstract] OR "Exercise"[MeSH Terms]) AND (("Cognitive Reserve"[MeSH Terms] OR "Cognitive Reserves"[Title/Abstract] OR "Cognitive Reserve"[Title/Abstract] OR "Brain Reserve"[Title/Abstract] OR "Brain Reserves"[Title/Abstract]) AND ((((((("severe Acquired Brain Injury"[Title/Abstract] OR "severe Acquired Brain Injuries"[Title/Abstract] OR "Stroke"[MeSH Terms] OR "Stroke"[Title/Abstract] OR "strokes"[Title/Abstract] OR "Cerebrovascular Accident"[Title/Abstract] OR "Cerebrovascular Accidents"[Title/Abstract] OR "Apoplexy"[Title/Abstract] OR "Brain Vascular Accident"[Title/Abstract] OR "Brain Vascular Accidents"[Title/Abstract]) NOT "Parkinson Disease"[MeSH Terms]) NOT "Parkinson"[Title/Abstract]) NOT "Multiple Sclerosis"[MeSH Terms]) NOT "multiple sclerosis, chronic progressive"[MeSH Terms]) NOT "multiple sclerosis, relapsing remitting"[MeSH Terms]) NOT "Multiple Sclerosis"[Title/Abstract])))
END (2004:2024[Jan 1, 2004: Jul 31, 2024])
N= 38
Google Scholar
cognitive reserve AND stroke OR post-stroke AND rehabilitation AND recovery OR cognitive recovery AND NOT traumatic -brain -injury -OR -multiple -sclerosis - OR -dementia - OR -Parkinson
N= 3700
Cochrane Library
cognitive reserve AND stroke AND rehabilitation AND recovery
N= 26
Scopus
TITLE-ABS-KEY ("cognitive reserve" AND stroke OR post-stroke AND rehabilitation AND recovery OR cognitive recovery AND NOT ("traumatic brain injury" OR "multiple sclerosis" OR dementia OR Parkinson))
N= 87
Please note that we used AND NOT to exclude articles on cognitive reserve and other diseases.
Finally, we updated Figure1, Table1, Table S1, and References section.

Round 2
Reviewer 1 Report
Comments and Suggestions for Authors
Thanks
Reviewer 3 Report
Comments and Suggestions for Authors
I thank the authors for clarifying my questions.
In my opinion it is a pity that you have decided to limit your review to social-behavior CR proxies but as long as it is clearly stated in the manuscript it looks quite clear and does not produce confusions.
There were a couple of mistakes along the text.
Line 62. Begin sentence with the capital letter. Better replace “because” with “Since”
Fig. 2. Check the legend. “Only” should not be capilatize. Check “emorragic”
Please, double-check your grammar along the text.